# Assessing the Impact of Simultaneous Co-Fermentation on Malolactic Bioconversion and the Quality of Cider Made with Low-Acidity Apples

Maria Luísa Cerri, Tatiane Aparecida Gomes [ID], Matheus de Melo Carraro, José Pedro Wojeicchowski, Ivo Mottin Demiate, Luiz Gustavo Lacerda, Aline Alberti [ID] and Alessandro Nogueira *[ID]

Graduate Program in Food Science and Technology, State University of Ponta Grossa, Av. Carlos Cavalcanti 4748, Uvaranas, Ponta Grossa CEP 84030-900, PR, Brazil; malucerri@yahoo.com.br (M.L.C.); tatianeapgomes@gmail.com (T.A.G.); mcarraro2000@gmail.com (M.d.M.C.); josepw@hotmail.com (J.P.W.); demiate@yahoo.com (I.M.D.); luizgustavo75@gmail.com (L.G.L.); alinealberti@gmail.com (A.A.)
* Correspondence: alessandronog@yahoo.com.br

**Abstract:** This study investigated the synergistic effects of combining *Saccharomyces cerevisiae* and *Oenococcus oeni* during the alcoholic fermentation of a low-acidity cider. The initial population of indigenous wild lactic acid bacteria (LAB) in the apple must was $10^4$ CFU/mL. Alcoholic fermentations were carried out without (Cider I) and with (Cider II) the *O. oeni* inoculation at $10^5$ CFU/mL. As *S. cerevisiae* grows, a declining trend was observed in indigenous and inoculated LAB populations. While the wild LAB exhibited higher sensitivity than *O. oeni*, they were not eliminated during alcoholic fermentation. The addition of *O. oeni* impacted the growth and metabolic activity of *S. cerevisiae*. The bioconversion of malic acid into lactic acid predominantly occurred during the growth phase (43–66%) and stationary phase (4–27%). The resurgence of *O. oeni* following alcoholic fermentation significantly impacted the production of volatile compounds. After 20 days of fermentation, Cider II displayed a twofold increase in these compounds, resulting in a more favorable sensory profile according to evaluators. Consequently, malolactic fermentation (MLF) coincided with alcoholic fermentation, leading to a reduction in malic acid content. Furthermore, post alcoholic fermentation, MLF positively enhanced the aromatic quality of low-acid cider made from apples with low acidity.

**Keywords:** malolactic fermentation; wild lactic acid bacteria; cider aroma; *O. oeni*; malic acid

## 1. Introduction

Cider, a sparkling fruity wine, is obtained through alcoholic and malolactic fermentations (MLFs) from apple must from dessert or industrial cultivars or their blends. MLF is mainly performed in dry ciders. With over 35 countries worldwide engaged in cider production [1,2], the industry is experiencing global growth in both small and large breweries. The allure of cider lies in its low alcohol content (2.0 to 7.0% *v*/*v*), gluten-free nature, and its appeal in terms of both functionality and sensory characteristics [2].

The quality of cider is inherently tied to the chemical composition of the raw materials, the intricacies of the processing technology applied, and the specific strains of microorganisms [3,4]. Cider fermentation is a complex biochemical process, marked by the conversion of sugars into ethanol and $CO_2$, and with the generation of aroma compounds, it is facilitated by cider-associated microbial communities, ultimately contributing to the complexity and diversity of the final product [5]. Notably, the apple cider fermentation process includes alcoholic fermentation (AF) and malolactic fermentation (MLF), both of which wield significant influence over cider quality [6].

Malic acid, the main organic acid in apple must, often constituting up to 90% of its composition, plays a pivotal role in shaping the sensory characteristics of cider. Nonetheless, the quantity and composition of organic acids are key determinants in achieving a

refreshing and harmonious cider profile. However, high levels of malic acid can render dry ($\leq$30 g/L of residual sugar) or demi-sec (30.1–50 g/L of residual sugar) cider rough and hard, negatively impacting its sensory quality [7,8]. Consequently, the regulation of malic acid concentration in cider holds significant importance. MLF, a secondary biological process conducted by lactic acid bacteria, is usually used for the de-acidification of wine and cider. It not only enhances microbial stability but also yields desirable flavor compounds [9]. Some authors argue that MLF is not a true fermentation process but rather an enzymatic reaction [10,11].

During MLF, the tart-tasting L-malic acid undergoes conversion into the milder L-lactic acid, along with the production of $CO_2$, catalyzed by malate lactase. This enzymatic reaction progresses slowly and is facilitated by bacteria from the genera *Oenococcus*, *Leuconostoc*, *Pediococcus*, and *Lactobacillus* [9]. Notably, *O. oeni* emerges as the predominant LAB strain (comprising over 50%) during the aging of ciders [8]. This strain offers several advantages, including high ethanol tolerance (above 13% *v/v*), resistance to $SO_2$, adaptability to low pH conditions, and minimal production of biogenic amine metabolites [12]. These qualities make *O. oeni* an ideal candidate for enhancing cider production, ensuring both quality and stability in the final product.

Therefore, lactic acid bacteria (LAB) play a crucial role in cider processing, and their introduction can occur naturally, sequentially, or simultaneously with the yeast inoculation. Simultaneous inoculation is emerging as the preferred method, with *O. oeni* exhibiting rapid growth during the yeast decline phase due to its better adaptation to the medium [6,13]. Research by Zhao et al. [14] has revealed that the introduction of malolactic fermentation results in the formation of higher alcohols, esters, and carbonyl compounds in ciders. Therefore, as already established in wine, controlled MLF in cider has the potential to enhance the complexity of the beverage's aroma and taste [6].

On the other hand, there is a lack of research assessing the impact of malolactic bioconversion on the quality of ciders made from apple must with low acidity levels (below 4.5 g/L). Therefore, this study aims to fill this research gap by evaluating the effects of malolactic bioconversion on cider production using dessert apples characterized by low acidity. The work will also investigate how this process influences the sensory attributes of the final beverage.

## 2. Materials and Methods

### 2.1. Materials

The 'Gala' and 'Fuji' apple cultivars were acquired from Boutin Agrícola in Porto Amazonas, Paraná, Brazil. Apples, amounting to 50 kg, were collected from six trees of the same variety and sampled at distinct cardinal points: the top, center, and bottom. To check the ripeness of these apples, the starch-iodine test was employed [15]. The results of this method indicated that the iodine value for the fruits fell within the range of 4–5, indicating their ripe stage.

The chemical standards used in the gas chromatography (GC) analysis were ethanal, ethyl propanoate, ethyl 3-methyl butanoate, propyl ethanoate, 2-methyl propyl ethanoate, ethyl butanoate, hexanone, 2-heptanone, 2-methyl-1-butanol, 2-hexanol, 2-octanone, ethyl hexanoate, purchased from Interchim (Montluçon, France), ethyl ethanoate, butyl ethanoate, 3-methyl butyl ethanoate, hexyl ethanoate, 2-hydroxy ethyl propanoate, 1-hexanol, ethyl octanoate, ethyl decanoate, butanoic acid, diethyl butanedioate, 2-phenyl-ethanol, ethyl dodecanoate, and octanoic acid, all acquired from Sigma–Aldrich (Steinheim, Germany), and 3-methyl-1-butanol was purchased from Merck (Darmstadt, Germany). All other reagents were of analytical grade.

### 2.2. Methods

#### 2.2.1. Washing and Sanitization of Apples

The objective of this experiment was to investigate the impact of washing and sanitizing apples on the population of wild LAB present in apple must. Gala and Fuji apples

were classified as industrial (exhibiting physiological and morphological defects, scars, and rot) and commercial (free from the aforementioned defects, falling into categories CAT 1 and CAT 2). The fruits were washed in potable water with mechanical agitation. Different concentrations of sodium hypochlorite (2.5% *m/v*) were tested, including 50, 100, 150, and 200 mg/L, with an immersion duration of 15 min at 4–5 °C. To prepare each treatment for 1 kg of fruit, different concentrations of sodium hypochlorite were diluted in 3 L of distilled water at a pH of 6.81. The sodium hypochlorite solution contained 84.8% of chlorine in HOCl form and 15.2% of chlorine in $OCl^-$ form. These sanitization processes were carried out at temperatures between 3 and 4 °C. Following the sanitization step, the fruits were rinsed in clean water (3 L) to remove any excess sanitizer. Subsequently, the apples were processed to extract apple must. It is important to note that all the experiments were performed in duplicate to ensure the reliability of the results.

2.2.2. Processing of Cider

The commercial apples were carefully selected, weighed and cleaned. Sanitization was not performed to avoid compromising the microbiota of wild LAB. The fruits were crushed and pressed (AGM Maquinas, Bento Gonçalvez, Brazil) at 294 kPa for 5 min, resulting in the extraction of integral apple must. The physical yield ($\eta_p$) averaged 71.5%. The apple musts were blended and depectinized using a pectinolytic enzyme (Pectinex-Batch® 1201371L, Novozymes Latin America LTDA, Araucária, Brazil) at 3.0 mL/hL at room temperature ($25 \pm 2$ °C) for 4 h. The depectinized apple must was racked, filtered in a line filter with a 0.5 μm opening (Hidro Filtros do Brasil®, Joinvile, Brazil) at 0.5 bar, and transferred to glass fermenters (Erlenmeyer) of 0.5 L with 85% loading, equipped with an airlock valve in aseptic conditions. Sulfite was not added to preserve the aromas [2]. Commercial inoculums of yeast *S. cerevisiae* (Fermol Aromê Plus®, AEB Group, Brescia, Italy), and lactic acid bacteria, *O. oeni* (Biolact Aclimatee 4R®, AEB Group, Brescia, Italy), both in active-dry form and preserved at 7 °C, were used. Two ciders were manufactured: Cider I with an inoculum of *S. cerevisiae* ($2.3 \times 10^5$ CFU/mL) and Cider II with an inoculum of *S. cerevisiae* ($2.1 \times 10^5$ CFU/mL) and *O. oeni* ($2.7 \times 10^5$ CFU/mL) at the same time.

Cider I was monitored over 10 days of alcoholic fermentation. The fermentation was stopped by centrifugation (Centrifuge Hitachi® CR21G2, High-Speed Refrigerated, Switzerland) at $7980 \times g$ at 4 °C for 15 min, aiming to avoid a spontaneous malolactic fermentation caused by indigenous lactic acid bacteria resistant to the sanitization process. Cider II was monitored for 20 days, and malolactic biotransformation was stopped by centrifugation. Then, the ciders without carbonation were bottled and stored at 2–3 °C (Freezer Bosch® do Brasil, model KDV47, São José dos Pinhais, Brazil) until further analysis. All the above operations were carried out in triplicate and performed under sterile conditions.

2.2.3. The Growth Curves of Yeast and LAB

Yeast and LAB growth were performed during batch fermentations to evaluate the microbial growth behavior in different inoculum conditions. The total yeast counting was performed in a culture medium of potato dextrose agar (PDA, Acumedia®, Lansing, MI, USA), hydrated and sterilized in an autoclave (Phoenix® VA 13811, Araraquara, São Paulo, Brazil) for 15 min (121 °C, 1.01 kgf/cm$^2$). Then, 10 mL of culture medium and 50 μL of 10% tartaric acid (Biotec®, Pinhais, Paraná, Brazil) were poured into 90 mm Petri dishes (J. Prolab®, São José dos Pinhais, Paraná, Brazil), and subsequently, the solid medium received 100 μL of the sample with scattering, using a sterile Drigalski spatula. The plates were then incubated at 26 °C for 24 h, with the subsequent counting of specific colonies. The results were expressed as CFU/mL.

The total LAB counting was performed in a selective medium prepared with 50 g of MRS agar (Lactobacilli MRS, Broth) diluted in 950 mL of distilled water, and the pH was adjusted to 4.8 with concentrated phosphoric acid (Biotec, Pinhais, Paraná, Brazil). A solution of antibiotics was prepared using 20 mg of actidione (Cycloheximide Sigma C1988-5G) and 5 mg of oxine (8-quinolinol Sigma H6878-25G) diluted in 10 mL of distilled

water and sterilized by filtering using a sterile membrane of 0.45 μm. A volume of sample (1 mL) was introduced into a Petri dish with 0.5 mL of antibiotic solution and 10 mL of the culture medium. The plaques were placed in anaerobic (Anaerobac, Probac do Brasil LTDA, São Paulo, Brazil) jugs (Permution, Curitiba, Brazil). After incubation for 8 days at 28 °C, the colonies were manually counted and expressed as CFU/mL.

### 2.2.4. Monitoring of Fermentation and Characterization of Ciders

The fermentation was monitored by the loss of mass from the system caused by the release of $CO_2$, and the weight was determined every two hours at a sensitivity of 0.001 g over 10 d (Cider I) and 20 d (Cider II) at 20 °C.

The determination of the sugar content was performed as described by Zielinski et al. [16] in an HPLC system (Waters Alliance 2695, Milford, MA, USA). The sugar was expressed as gram per L of cider. The ethyl alcohol content was determined by ebulliometer (accuracy of 0.1% vol.) and expressed °GL. The L-malic, D-lactic, and L-lactic acids were quantified by enzyme kit (L-malic acid, cod. EZA786, and D-lactic, cod. EZA889, L-lactic, cod. EZA890, Enzyplus®, Raisio Diagnostics Spa, Rome, Italy). The total acidity was determined by neutralization with NaOH 0.1 N, using phenolphthalein as an indicator and expressed as malic acid, and the volatile acidity was calculated as acetic acid. Both were expressed as g/L [17]. The pH was determined using a pHmeter (pH digital micro process Tecnal®, model TEC3-MP, Piracicaba, São Paulo, Brazil). All the analyses were performed in triplicate.

### 2.2.5. Analysis of Volatile Compounds

The capture of compounds using headspace was performed according to the method of Saerens et al. [18] with modifications. Samples (6 mL) of apple juice and ciders were placed in glass vials with a capacity of 20 mL. Then, 50 μL of internal standard (Heptanoic acid, Merck®) was added. Prior to analysis, the samples remained for 10 min at 60 °C under agitation in the oven of the automatic injector (Young Lin Instrument® Gas Chromatograph, Anyang, Republic of Korea).

The analysis of aromatic compounds was performed by gas chromatography according to the method of Pietrowski et al. [19]. The compounds were identified by comparing retention times with those obtained in the reference solution. Equation (1) was used to quantify the concentrations of identified volatile compounds, where *C* is the concentration of the component (mg/L), *A* is the concentration of the substance in the reference solution (mg/L), *h* is the peak area of the substance in the sample, *H* is the peak area of substance in the reference, *I* is the peak area of internal standard in the reference, and *i* is the peak area of internal standard in the sample.

$$C = A \times (h/H) \times (I/i) \tag{1}$$

### 2.2.6. Sensory Analysis

The triangle test—a preference test with a nine-point hedonic scale from 1 = "I disliked extremely" to 9 = "I liked extremely"—was performed, and purchase intention was measured according to Dutcosky [20], after approval by the Research Ethics Committee (COEP) of the State University of Ponta Grossa (UEPG); project number 62047516.3.0000.0105.

Untrained panelists (n = 104) older than 18 years, mostly women (58%), participated in the sensorial analysis. The samples were served to panelists in disposable cups (50 mL) of opaque white color (Copobrás®, Guarulhos, São Paulo, Brazil), in a volume of 20 mL per person at 10 °C. The panelists were in individual cabins to prevent the exchange of information. For the triangle test, each panelist received three coded samples with random numbers: two samples of cider obtained by alcoholic fermentation (Cider I) and one sample of cider obtained by alcoholic and malolactic fermentation (Cider II). The panelists answered open questions about what they knew about cider, frequency of drinking, and preference between dry or soft fermented beverages. Moreover, preference and purchase

intention tests were also investigated. The samples were served following the same procedures already described. The results of the triangle test were evaluated by the significance table of the triangle test (P = 1/3), and the Student's test (T) was used as a preference test, according to Dutcosky [20].

### 2.2.7. Statistical Analysis

The data were shown as average and standard deviation. The homogeneity of variance was verified by Levene's test or the F-test ($p \geq 0.05$). The differences between the samples were evaluated using one-way ANOVA, followed by Fisher's LSD test. The statistical analysis was performed using Statistica v. 13.3 software (TIBCO Software Inc., Palo Alto, CA, USA) and Origin® 9.0 (OriginLab, Northampton, MA, USA).

## 3. Results and Discussion

### 3.1. Wild LAB in the Apple Must

The use of industrial apples or fruits still classified as commercial quality, as well as washing and sanitizing operations, are common practices in the cider industry. The indigenous microbiota are mainly present on the surface of apples and cider production equipment [21]. In this study, we assessed the impact of washing and sanitizing apples, categorized as either commercial or industrial, on the population of wild lactic acid bacteria in apple must (Table 1). The industrial apples exhibited a notably higher LAB population, with the approximate difference of 2 logs, reaching $3.7 \times 10^6$ CFU/mL. This disparity can be attributed to a higher prevalence of physical injuries and phytopathological defects in the industrial apple samples. Washing in potable water with mechanical agitation proved effective, resulting in a reduction of 1 log for commercial apples and up to 3 logs for industrial apples in the must.

**Table 1.** Effect of apple treatments on the population (CFU/mL) of wild lactic acid bacteria in apple must.

| Apples | Without Wash | | After Wash | | Sodium Hypochlorite Content * (mg/L) | | | | | | | |
| | | | | | 50 | | 100 | | 150 | | 200 | |
| | Min | Max | Min | Max | Min | Max | Min | Max | Min | Max | Min | Max |
| Commercial | $2.7 \times 10^3$ | $1.5 \times 10^5$ | $1.7 \times 10^2$ | $7.7 \times 10^3$ | $3.4 \times 10^2$ | $4.1 \times 10^4$ | $<4.0 \times 10^1$ | $6.1 \times 10^2$ | $2.6 \times 10^2$ | $4.0 \times 10^3$ | $1.5 \times 10^3$ | $6.1 \times 10^3$ |
| Industrial | $6.2 \times 10^5$ | $3.7 \times 10^6$ | $5.0 \times 10^2$ | $1.1 \times 10^4$ | $2.6 \times 10^2$ | $6.8 \times 10^2$ | $6.6 \times 10^2$ | $3.7 \times 10^3$ | $<4.0 \times 10^1$ | $1.0 \times 10^4$ | $<4.0 \times 10^1$ | $1.4 \times 10^3$ |

Note: * after 15 min at 3–4 °C.

When fruits were washed and sanitized, the results were most pronounced with sanitizer concentrations greater than 50 mg/L of active chlorine. At concentrations ranging from 100 to 200 mg/L of active chlorine, LAB populations in the apple must range from <40 to $10^4$ CFU/mL. However, the results were not significantly influenced by the sanitizer concentration (Table 1). This effect might be attributed to the initial population's variability the diversity of LAB strains, and potential protection against the effect of the sanitizer in regions of the peduncle, calyx, and epicarp lesions. This study indicates that the wild LAB population may exhibit great natural variability and may still be affected by washing and sanitation practices. Therefore, to achieve a satisfactory effect of MLF, a culture LAB must be inoculated into the must or cider at the end of alcoholic fermentation.

### 3.2. Kinetics of Lactic Acid Bacteria in Cider Fermentation

The initial LAB populations were $2.8 \times 10^4$ CFU/mL (Cider I: wild LAB) and $2.7 \times 10^5$ CFU/mL (Cider II: *O. oeni*). Both ciders had the same yeast inoculum, with a population of $1.2 \times 10^5$ CFU/mL (Table 2).

**Table 2.** Chemical composition of apple must and ciders.

| Analytical Parameters (g/L) | Apple Must | Ciders | |
|---|---|---|---|
| | | I * | II ** |
| Total sugars [1] | 118.95 ± 0.24 | 16.45 [a] ± 0.19 | 2.27 [b] ± 0.01 |
| Ethanol (%) | nd | 5.7 [b] ± 0.1 | 6.4 [a] ± 0.1 |
| Total acidity [2] | 2.43 ± 0.04 | 2.19 ± 0.01 | 2.74 ± 0.04 |
| Malic acid | 2.25 ± 0.15 | 0.24 [a] ± 0.08 | 0.15 [b] ± 0.05 |
| L-lactic acid | nd | 0.73 ± 0.04 | 0.85 ± 0.06 |
| D-lactic acid | nd | 0.69 ± 0.06 | 0.40 ± 0.04 |
| Volatile acidity [3] | 0.66 ± 0.10 | 1.37 ± 0.08 | 1.42 ± 0.12 |
| pH | 3.84 ± 0.02 | 3.99 ± 0.02 | 4.10 ± 0.01 |
| Yeast, UFC/mL | nd | $2.8 \times 10^8$ | $8.5 \times 10^5$ |
| LAB, UFC/mL | $2.8 \times 10^4$ | $1.6 \times 10^2$ | $3.8 \times 10^7$ |

Note: (*) fermentation stopped after 10 days, without wild LAB growth. (**) fermentation stopped. after 20 days, with the growth of *O. oeni* after alcoholic fermentation. [1] sum of glucose, fructose and sucrose. [2] g/L of malic acid. [3] g/L of acetic acid. (nd) not detected. Letters on the same line indicate a significant difference ($p < 0.05$).

Figure 1A illustrates that both wild and inoculated LAB tend to reduce their populations as *S. cerevisiae* grows. Wild LABs proved to be more sensitive than *O. oeni* but were not eliminated from the medium. This effect persisted until the fifth day, after which the population remained stable, ranging between $10^3$ and $10^4$ CFU/mL, until the tenth day of alcoholic fermentation. This initial reduction in the LAB population during alcoholic fermentation has also been noted in prior studies by Saguir et al. [11] and Dierings et al. [13]. However, the presence of *O. oeni* in a similar population to that of wild LAB impacted the growth of *S. cerevisiae* (Figure 1A), subsequently affecting the rate of sugar consumption and ethanol production (Figure 1B). This impact is likely due to competition for nutrients and release of yeast-inhibiting compounds, such as acetic acid [6].

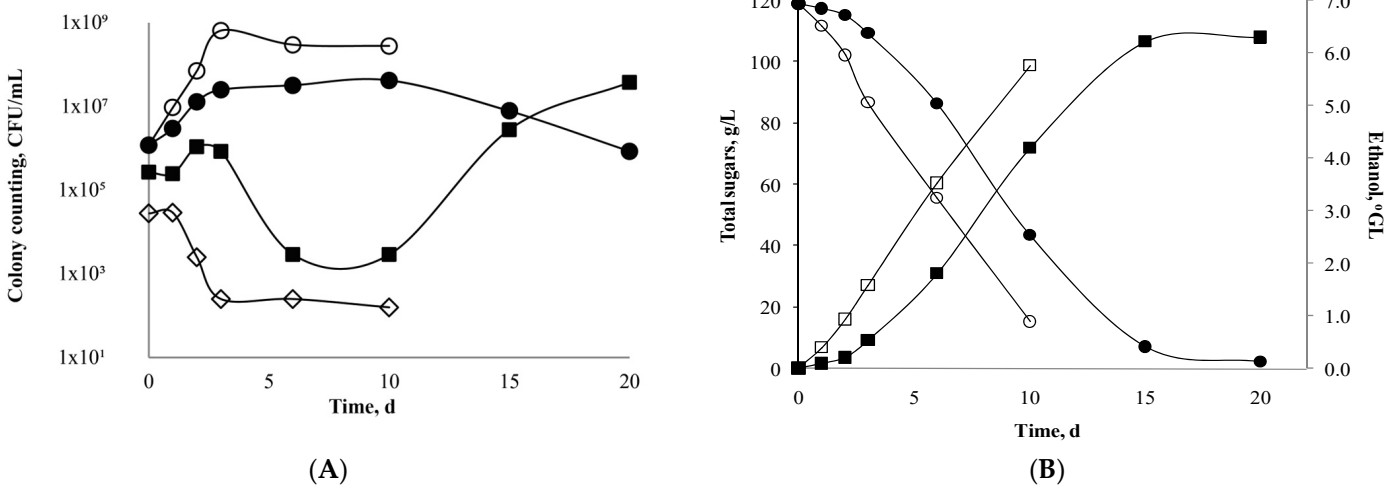

**Figure 1.** Rate of growth (**A**) of yeast (○, ●) and lactic acid bacteria (◇ wild LAB; ■ *O. oeni*); (**B**) sugar evolution (●, ○) and ethanol production (■; ◇). Note: Empty symbols "Cider I (10 days of fermentation)" and filled "Cider II (20 days of fermentation)".

Table 2 shows a reduction in total sugars during the alcoholic fermentation of apple must, amounting to 86.2 for Cider I and 98.1% for Cider II, with a direct impact on the ethanol content. These effects are related to the alcoholic fermentation time. Nevertheless, these values are consistent with values found in the literature for ciders made with dessert apples [13].

To prevent the natural growth of lactic acid bacteria, the alcoholic fermentation of the cider without LAB inoculum was interrupted on the tenth day. Dierings et al. [13] observed that the wild LAB population exceeded $10^6$ CFU/mL after 10 days of cider alcoholic fermentation. In the yeast decline phase, commercial LAB (*O. oeni*) exhibited rapid growth

beyond the tenth day, reaching $3.8 \times 10^7$ CFU/mL (Figure 1A), without hindering the ethanol content (Figure 1B). This fact can be attributed to the strain's adaptation to the cider medium and the release of nutrients, including amino acids, peptides, minerals, and organic acids through yeast autolysis [22,23].

### 3.3. Evolution of Organic Acids

Table 2 provides an overview of the organic acid and pH values in apple must and ciders after 10 days (Cider I) and 20 days (Cider II) of fermentation. In Figure 2, the initial three days of alcoholic fermentation showed notable changes in pH, total titratable acidity, volatile acidity, and organic acids. After this initial period, these parameters exhibited minimal modifications in the subsequent stages of fermentation. Total titratable acidity and organic acids remained relatively stable, as did pH and volatile acidity, as shown in Figure 2. This behavior was observed in both ciders. Regarding total titratable acidity, an initial increase was observed within the first day, followed by a sharp decline (40 to 43%, Cider I and II, respectively) and a subsequent slight increase during the alcoholic fermentation phase. As expected, titratable acidities decreased, and pH values increased during MLF, consistent with findings in the study of Reuss et al. [7]. Note that *O. oeni* can transform malic acid into lactic or acetic acid in the process of obtaining energy to grow. Wild LAB showed a similar metabolic behavior (Figure 2A,C).

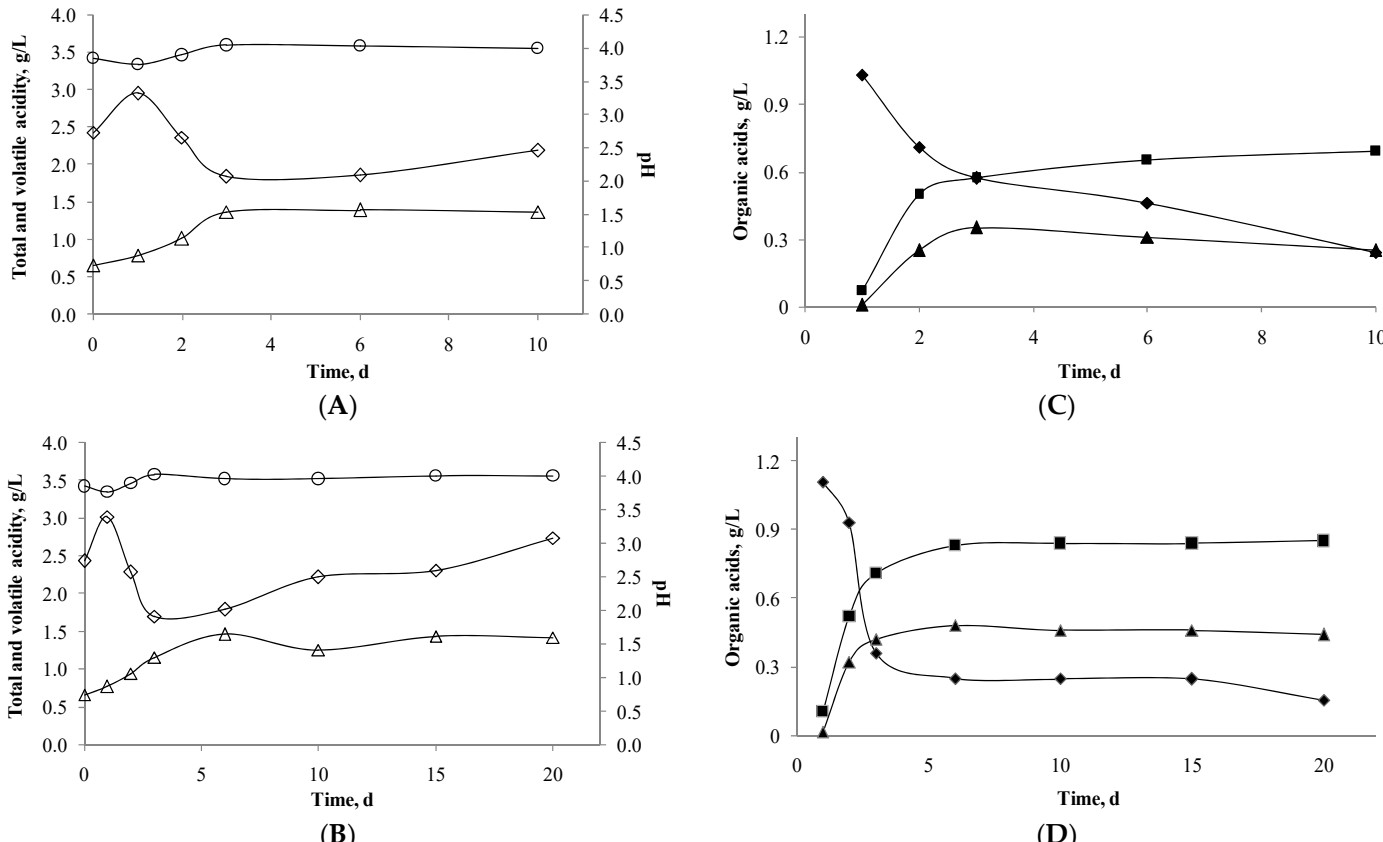

**Figure 2.** Evolution of total acidity (◇), volatile acidity (△), and pH (○); and organic acids L-malic acid (◆); L-lactic acid (■), and D-lactic acid (▲), during cider fermentation. Note: (**A,C**) corresponds to alcoholic fermentation with yeast inoculum over 10 days, Cider I. (**B,D**) corresponds to alcoholic fermentation with yeast and lactic bacteria inoculum over 20 days, Cider II.

After the initial three days of alcoholic fermentation, which includes the yeast growth phase, malic acid content decreased by 43 and 66% for Cider I and II, respectively. A more significant reduction was observed in Cider II, attributed to the inoculation of *O. oeni*. Over 10 days, malic acid content in both ciders decreased by approximately 70%. This means there was a rapid reduction in malic acid during the early stages of fermentation, with the biotransformation of malic acid into the L and D lactic acid isomers continuing at a slower pace during the stationary phase (Figure 2C,D). For Cider I, MLF resulted in 0.70 of L-lactic acid and 0.32 g/L of D-lactic acid, while Cider II produced 0.87 g/L of L-lactic acid and 0.5 g/L of D-lactic acid. At the end of 10 and 20 days of fermentation, the residual content of malic acid was 0.3 and 0.15 g/L for Cider I and II, respectively, corresponding to MLF percentages of 70 and 85%, for the two ciders (Table 2). The regrowth of *O. oeni* after 10 days of fermentation (Figure 1A) did not significantly impact the progress of MLF. Therefore, in the production of ciders using dessert apples with low acidity, a substantial portion of malolactic biotransformation occurs during the growth phase of microorganisms, both in wild LAB and in inoculated LAB. Furthermore, the results indicate that malic acid biotransformation may not reach completion in both ciders. Alberti and Nogueira [2] described that MLF typically takes place after alcoholic fermentation and may last for 20–30 days during storage. However, under the conditions of this study, it appears that an extended period of MLF may not be necessary, or perhaps is not cost-effective, in terms of cost and benefit.

### 3.4. Synthesis of Volatile Compounds

Additionally, this study revealed interesting insights into the profile of volatile compounds in ciders (Table 3). In both ciders, all volatile compounds exceeded their respective sensory thresholds, underscoring the significance of these compounds in shaping the aroma of ciders. During the first 10 days of alcoholic fermentation, no sensory differences were observed between the ciders. However, a substantial discrepancy emerged between Cider I and Cider II after 20 days of fermentation, with the sum of identified volatiles being approximately twice as small in Cider I. It is worth noting that Zhao et al. [14] reported similar volatile contents for cider with and without malolactic fermentation. The duration of MLF time and the regrowth of LAB after 10 days of alcoholic fermentation (Figure 1A) might have influenced the synthesis of volatile compounds in Cider II (Table 3). The identified and quantified volatile compounds belong to the esters and alcohols families, which, according to Wang et al. [24], contribute fruity aromas to the cider. Some of these compounds were synthesized up to four times more in Cider II (Table 3). For instance, Sánchez et al. [25] highlighted the significance of higher alcohols like 3-methyl-1-butanol and 2-phenyl ethanol in traditional and commercial ciders due to their aroma intensity and high variation. These volatile compounds are known for imparting fruity and rosy, as well as fruity and honey-like, aromas, respectively [26]. In this study, both compounds were found in their highest concentrations in Cider II, suggesting they may have been synthesized by *O. oeni*. However, it is important to consider that the influence of *O. oeni* on the yeast population may have also contributed to the production of these compounds, given the slower rate of alcoholic fermentation [2].

**Table 3.** Volatile compounds in apple must and cider elaborated in the presence and absence of malolactic fermentation.

| Volatile Compounds (mg/L) | Apple Must | Ciders | | |
|---|---|---|---|---|
| | | I * | II ** | Threshold |
| **Ester** | | | | |
| Ethyl ethanoate | 3.79 ± 1.93 | 140.62 [b] ± 0.82 | 442.56 [a] ± 4.94 | 7.5 [(1;2;3;6)] |
| Ethyl butanoate | nd | 1.14 [a] ± 0.79 | 2.76 [a] ± 0.72 | 0.02 [(1;2;3)] |
| Isopentyl acetate | 0.39 ± 0.19 | 12.14 [b] ± 0.89 | 19.55 [a] ± 0.78 | 0.03 [(1;2;3)] |
| Hexyl ethanoate | 0.11 ± 0.09 | 1.15 [a] ± 0.93 | 1.71 [a] ± 0.84 | 0.67 [(2;3;6)] |
| 2-hydroxy ethyl propanoate | 44.39 ± 6.50 | 36.87 [b] ± 5.39 | 123.27 [a] ± 3.46 | 1.8 [(3)] |
| Ethyl octanoate | nd | 10.56 [b] ± 1.07 | 17.03 [a] ± 1.35 | 0.002 [(1;6)] |
| Ethyl decanoate | 0.04 ± 0.02 | 6.78 [b] ± 1.75 | 7.20 [a] ± 0.97 | 0.5 [(1;6)] |
| 1,4-ethyl butanoate | 6.46 ± 7.06 | 14.78 [a] ± 0.21 | nd | 0.02 [(1;2;3)] |
| Ethyl dodecanoate | 3.74 ± 1.08 | 4.77 [b] ± 0.01 | 11.43 [a] ± 0.25 | 0.5 [(1)] |
| **Aldehyde** | | | | |
| Acetaldehyde (ethanal) | 1.54 ± 1.38 | 12.78 [a] ± 0.82 | nd | 0.5 [(3)] |
| **Acid** | | | | |
| Butanoic acid | nd | 47.11 [b] ± 3.68 | 53.57 [a] ± 3.19 | 0.24 [(5)] |
| Octanoic acid | 2.22 ± 1.44 | 12.77 [b] ± 0.01 | 23.25 [a] ± 0.53 | 10 [(1;6)] |
| **Higher alcohols** | | | | |
| 2-phenyl-ethanol | 11.22 ± 3.19 | 27.95 [b] ± 2.27 | 61.83 [a] ± 0.15 | 10 [(1;2;3)] |
| 3-methyl-1-butanol | 2.37 ± 1.19 | 187.54 [b] ± 6.51 | 351.36 [a] ± 1.46 | 30 [(1;2;3)] |
| 2-hexanol | nd | 16.72 [b] ± 1.40 | 25.86 [a] ± 6.33 | 15 [(3)] |
| **Acetone** | | | | |
| 2-heptanone | 0.07 ± 0.04 | nd | 3.22 ± 0.67 | 0.0082 [(4)] |
| **∑ of compounds** | **76.34** | **533.68** | **1144.6** | **-** |

Note: (*) 10 days of fermentation; (**) 20 days of fermentation. nd = not detected; [a,b] = different letters in the same line indicate that there is a significant difference between concentrations of samples ($p < 0.05$); [(1)] Yu et al. [27]; [(2)] Li et al. [6]; [(3)] Wei et al. [28]; [(4)] Karl et al. [29]; [(5)] Arcari et al. [30]; [(6)] Baiano et al. [31].

Moreover, compounds such as ethyl decanoate, ethyl dodecanoate, butanoic acid, and 2-phenyl ethanol compounds were identified in both ciders, with a significant difference in concentration, which was notably higher in Cider II. These compounds are associated with sensorial notes of flowers and fruits in ciders, as observed by Rosend et al. [32] and Villière et al. [33].

### 3.5. Sensorial Evaluation

In the context of the triangular test involving untrained tasters (n = 104), it was found that 39% of them said they recognized cider as an "alcoholic fermented apple beverage", while 35% knew cider as an "alcoholic fermented fruit beverage" and 26% had no prior knowledge of cider. Additionally, 65% of the tasters reported consuming wine and/or fermented fruit. Within this group, 9% indicated they consumed these beverages weekly, 21% monthly, and 35% only on special occasions, such as Christmas and New Year's Eve. This preference for consuming wine and fermented fruit beverages on special occasions could be attributed to the perceived low sensory quality of commercial ciders [2].

The triangle test was employed to assess whether the panelists could differentiate the effects of malolactic biotransformation in Cider II. Of the 104 untrained panelists, 65 individuals (62.5%) were able to identify the sample with malolactic biotransformation. The test results revealed a significant difference between the samples, with a significance level of 95% ($p < 0.05$). Therefore, it is concluded that the evaluators sensorially perceived the difference caused by *O. oeni* in the profile of the cider and that the ciders were considered different from each other.

The difference in volatile compounds (Table 3) led to Cider II (more fruited) being perceived as 9% "more sensorily acceptable" and 9% "more likely to be purchased" than Cider I (Table 4). This finding is particularly significant as it emerged from untrained tasters with limited prior knowledge of the beverage.

**Table 4.** Responses of hedonic scale in the Preference Test and of grade points in the Purchase Intention Test of cider samples obtained by presence (I) and absence (II) *O. oeni* inoculum.

| Hedonic Scale/Categories | | Ciders | |
|---|---|---|---|
| | | I * | II ** |
| Preference Test | I dislike extremely | 10 | 2 |
| | I dislike very much | 7 | 5 |
| | I dislike moderately | 5 | 8 |
| | I dislike slightly | 8 | 9 |
| | I neither like nor dislike | 12 | 9 |
| | I like slightly | 27 | 27 |
| | I like moderately | 21 | 27 |
| | I like very much | 11 | 16 |
| | I like extremely | 3 | 1 |
| Acceptance Index ($p$ = 0.193) | | 59.6 | 68.3 |
| Purchase Intention Test | Definitely wouldn't buy | 13 | 9 |
| | Probably wouldn't buy | 26 | 18 |
| | Might would buy/Might wouldn't buy | 39 | 49 |
| | Probably would buy | 23 | 19 |
| | Definitely would buy | 3 | 9 |
| Purchase intent % | | 62.0 | 71.0 |

Note: (*) 10 days of fermentation; (**) 20 days of fermentation.

Thus, this demonstrates that the impact of MLF extends beyond the mere biotransformation of malic acid; it significantly influences the formation of aromatic compounds that enhance the overall quality of cider produced from low-acidity apples.

## 4. Conclusions

The washing and sanitizing (sodium hypochlorite > 100 mg/L) of apples can effectively reduce wild LAB populations, although complete elimination is not achieved. The inoculum of *O. oeni* had a notable impact on the growth of *S. cerevisiae*. During the yeast growth phase, malic acid content was reduced by 43 and 66% for Cider I (wild LAB) and II (inoculated LAB), respectively, and at the end of alcoholic fermentation, this reached 70% in both ciders. Notably, during the yeast decline phase, the regrowth of *O. oeni* did not affect the bioconversion of malic acid to L-D lactic. Furthermore, after 20 days of alcoholic fermentation, it was found that the production of volatile compounds in Cider II was approximately twice as high as in Cider I. This included the synthesis of various esters and alcohol compounds that contribute fruity aromas to the cider, with some compounds being produced up to four times more in Cider II. As a result, Cider II was preferred by the tasters due to its enhanced sensory profile. In the context of this study, which primarily involved low-acid apples, the findings suggest that an extended period of more than 20 days for MLF may not be necessary and might not be justifiable from an economic standpoint in terms of cost and benefit.

**Author Contributions:** M.L.C.: formal analysis, writing (original draft); T.A.G.: formal analysis, writing (original draft); M.d.M.C.: formal analysis, writing (original draft); J.P.W.: writing (original draft); I.M.D.: methodology, writing (review and editing); L.G.L.: methodology, writing (review and editing); A.A.: supervision, methodology, writing (review and editing); A.N.: resources, project administration, funding acquisition, writing (review and editing). All authors have read and agreed to the published version of the manuscript.

**Funding:** This research was funded by the National Council for Scientific and Technological Development (CNPq) [grant No. 313417/2019-9], the Araucária Foundation (FA), the Coordenação de Aperfeiçoamento de Pessoal de Nível Superior-Brasil (CAPES) [grant finance code 001].

**Institutional Review Board Statement:** The study was approved by the Research Ethics Committee (COEP) of the State University of Ponta Grossa (UEPG); project number 62047516.3.0000.0105.

**Informed Consent Statement:** Informed consent was obtained from all subjects involved in the study.

**Data Availability Statement:** Data are unavailable due to privacy restrictions. Researchers can provide if necessary.

**Conflicts of Interest:** The authors declare no conflict of interest.

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
