# Peer review of "Assessing the Impact of Simultaneous Co-Fermentation on Malolactic Bioconversion and the Quality of Cider Made with Low-Acidity Apples"

_fermentation, doi:10.3390/fermentation9121017_

Round 1
Reviewer 1 Report
Comments and Suggestions for Authors
Line 29. The performance of malolactic fermentation is necessary? Occurs every time? Or as in the case of grape wine sometimes occurs and sometimes not.
Lines 36-37. Which alcohols? Ethanol is mainly produced. The other in considerable lower content. Please revise.
Line 44. Please add in parenthesis the sugar content of dry and demi-sec cider since there are numerous different regulations in the world.
Line 55. Please use SI system only.
Line 108. Fermenters 0.5L? Conical flasks? Please provide details.
Line 152. “ebulliometer”. Please provide the accuracy of that.
Table 2. Please provide all the necessary details. The table should be able to stand alone. For example acidity as g/L of what? volatile acidity as g/L of what?
No selective medium for Oeni was used. How the authors are sure that only that microorganism survived?
Please explain why to stop fermentation at 10 day in sample 1 and 20 days at sample 2. Why not to compare the malolactic fermentation of wild LAB?
Figure 2. The axis of pH is not appropriate. In the present form no difference can be observed. Please use an axis with pH 3 to 4 for example.
Problems with the design of the study:
1. The first experiment with washing and sanitation seems a little bit with no correlation with the rest of the study.
2. No selective medium for Oeni.
3. Different fermentation times.
4. The authors measured and compared completely different products. Why? In my opinion they should either stop the fermentation in both samples at 10 or 20 days. The fermentation process with the wild LAB was better if someone compare the 10 days of fermentation (higher ethanol, lower sugars, etc.
5. Comparing the volatile/aroma compounds in different parts of fermentation is completely wrong (10 days for I and 20 days for II). The same applies for sensory analysis.
It would be very interesting to see the result of wild LAB fermentation after 20 days and its comparison with Oeni. It is difficult to understand the importance of the present study and its results.
Comments on the Quality of English LanguageMinor corrections are needed.
Author Response
Manuscript ID: fermentation -2728983
Title: Assessing the Impact of Simultaneous Co-Fermentation on Malolactic Bioconversion and in the Quality of Cider Made with Low-acidity Apples
Dear Dr. Ms. Yulia Tang
Fermentation
Thank you for your response to our manuscript fermentation-2728983.
The constructive criticism of the reviewers was much appreciated and we revised our manuscript accordingly. All suggestions were accepted. More information and details were included in the text, and the manuscript was revised thoroughly. All the modifications performed in the revised manuscript are highlighted. The accompanying document at the bottom of this letter contains a point-to-point reply to the reviewer.
Thank you in advance for your attention.
Reviewer # 1
- Line 29. The performance of malolactic fermentation is necessary? Occurs every time? Or as in the case of grape wine sometimes it occurs and sometimes not.
Answer: Thanks for the comment. The MLF is mainly performed in dry ciders. The changes were performed accordingly.
-Lines 36-37. Which alcohols? Ethanol is mainly produced. The other in considerably lower content. Please revise.
Answer: Thank you for your comments. The suggestion was accepted and changes were performed accordingly. Please check the information in the text.
-Line 44. Please add in parenthesis the sugar content of dry and demi-sec cider since there are numerous different regulations in the world.
Answer: Thank you for your comments. The suggestion was accepted and changes were performed accordingly. Please check the information in the text.
-Line 55. Please use SI system only.
Answer: Thank you for your comments. The suggestion was accepted and changes were performed accordingly. Please check the information in the text.
-Line 108. Fermenters 0.5L? Conical flasks? Please provide details.
Answer: Thank you for your comments. The suggestion was accepted and changes were performed accordingly. Please check the information in the text.
-Line 152. “ebulliometer”. Please provide the accuracy of that.
Answer: Thank you for your comments. The suggestion was accepted and changes were performed accordingly. Please check the information in the text.
Table 2. Please provide all the necessary details. The table should be able to stand alone. For example, acidity as g/L of what? volatile acidity as g/L of what?
Answer: Thank you for your comments. The suggestion was accepted and changes were performed accordingly. Please check the information in Table 2.
No selective medium for Oeni was used. How the authors are sure that only that microorganism survived?
Answer: Thanks for the question. In previous tests and works such as Dierings et al. (2013), it was observed that the growth curve of O. oeni after alcoholic fermentation showed a different behavior than of wild LABs. Thus, the O. oeni cells that survive show a better adaptation to cider, resulting in rapid growth in the decline phase of S. cerevisiae.
Please explain why to stop fermentation at 10 days in sample 1 and 20 days at sample 2. Why not to compare the malolactic fermentation of wild LAB?
Answer: In this work, we would like to present the effect of MLF on the quality of cider with a simultaneous inoculum of S. cerevisiae and O. oeni. We needed to compare it with cider without the inoculum and the malolactic effect. However, we agree that a third malolactic cider through wild LAB could be interesting, but we believe the result would be similar.
Figure 2. The axis of pH is not appropriate. In the present form no difference can be observed. Please use an axis with pH 3 to 4 for example.
Answer: Ok, we modified the pH axis in Fig. 2. "Nevertheless, we contend that by doing so, we are compelled to highlight a potential impact on the curve that, in reality, did not occur." In discussion with other co-authors, we thought it was better to keep the original format.
Problems with the design of the study:
- The first experiment with washing and sanitation seems a little bit with no correlation with the rest of the study.
Answer: Thanks for the suggestion. The correlation between the first experiment and the others was improved. Please check lines 199-200 and 291-221.
- No selective medium for Oeni.
Answer: Thanks for the question. In previous tests and works such as Dierings et al. (2013), it was observed that the growth curve of O. oeni after alcoholic fermentation showed a different behavior than of wild LABs. Thus, the O. oeni cells that survive show a better adaptation to cider, resulting in rapid growth in the decline phase of S. cerevisiae.
- Different fermentation times.
Answer: Our objective was to verify the effect of malolactic during alcoholic fermentation and the impact it would have on a cider without malolactic and another with malolactic. In some countries, the malolactic period, after sugar exhaustion by yeast, is 20-30 days, which is why we use different times.
- The authors measured and compared completely different products. Why? In my opinion, they should either stop the fermentation in both samples at 10 or 20 days. The fermentation process with the wild LAB was better if someone compared the 10 days of fermentation (higher ethanol, lower sugars, etc.
Answer: Thanks for the suggestion. Similar answer to item 3. Furthermore, even experimenting as suggested, the results would be the same.
- Comparing the volatile/aroma compounds in different parts of fermentation is completely wrong (10 days for I and 20 days for II). The same applies for sensory analysis.
Answer: Similar answer to item 3 and 4.
We think we were able to respond adequately to all issues raised by the reviewer and hope you will find our manuscript now acceptable for publication in Fermentation. If any further modifications are necessary, please inform us, and we will make every effort to implement them.
On behalf of all authors,
Yours sincerely,
Prof. Alessandro Nogueira,
State University of Ponta Grossa
Reviewer 2 Report
Comments and Suggestions for Authors
The paper is well written and represents and advance in the research of apple fermentation for cider production. The impact of co-inoculation of O.oeni with yeats, in the must, considerable improve the volatile compound and acceptability of beverage. However, I don´t know which type of apple using for fermentation assay (from table 1). The secction 2.2.1 has not relevance for the manuscript and could be deleted or add more information about the relation between the sanitization of apple and cider production.
Author Response
Manuscript ID: fermentation -2728983
Title: Assessing the Impact of Simultaneous Co-Fermentation on Malolactic Bioconversion and in the Quality of Cider Made with Low-acidity Apples
Dear Dr. Ms. Yulia Tang
Fermentation
Thank you for your response to our manuscript fermentation-2728983.
The constructive criticism of the reviewers was much appreciated and we revised our manuscript accordingly. All suggestions were accepted. More information and details were included in the text, and the manuscript was revised thoroughly. All the modifications performed in the revised manuscript are highlighted. The accompanying document at the bottom of this letter contains a point-to-point reply to the reviewer.
Thank you in advance for your attention.
Reviewer # 2
The paper is well written and represents and advance in the research of apple fermentation for cider production. The impact of co-inoculation of O.oeni with yeats, in the must, considerable improve the volatile compound and acceptability of beverage. However, I don´t know which type of apple using for fermentation assay (from table 1).
Answer 1: Thank you for your comments. The apple used was classified as commercial. The information was corrected. Please check the text.
The section 2.2.1 has not relevance for the manuscript and could be deleted or add more information about the relation between the sanitization of apple and cider production
Answer 2: Thanks for the suggestion. The correlation between the first experiment and the others was improved. Please check lines 199-200 and 291-221.
We think we were able to respond adequately to all issues raised by the reviewer and hope you will find our manuscript now acceptable for publication in Fermentation. If any further modifications are necessary, please inform us, and we will make every effort to implement them.
On behalf of all authors,
Yours sincerely,
Prof. Alessandro Nogueira,
State University of Ponta Grossa
Reviewer 3 Report
Comments and Suggestions for Authors
The authors report on the influence of simultaneous co-fermentation inoculated with Saccharomyces cerevisiae and Oenococcus oeni on the malolactic bioconversion and the quality of cider. The study provides a way of producing low acidity cider, which has practical value in the field of fermented beverages. See the attachment for detailed suggestions.

Author Response
Manuscript ID: fermentation -2728983
Title: Assessing the Impact of Simultaneous Co-Fermentation on Malolactic Bioconversion and in the Quality of Cider Made with Low-acidity Apples
Dear Dr. Ms. Yulia Tang
Fermentation
Thank you for your response to our manuscript fermentation-2728983.
The constructive criticism of the reviewers was much appreciated and we revised our manuscript accordingly. All suggestions were accepted. More information and details were included in the text, and the manuscript was revised thoroughly. All the modifications performed in the revised manuscript are highlighted. The accompanying document at the bottom of this letter contains a point-to-point reply to the reviewer.
Thank you in advance for your attention.
Reviewer # 3
The authors report on the influence of simultaneous co-fermentation inoculated with Saccharomyces cerevisiae and Oenococcus oeni on the malolactic bioconversion and the quality of cider. The study provides a way of producing low-acidity cider, which has practical value in the field of fermented beverages. See the attachment for detailed suggestions.
Answer: Thank you for your comments.
Line 113-114: Why is the yeast inoculation amount inconsistent between CiderI and CiderII?
Answer: Thank you for your comments. The values were checked and an average between the initial yeast populations was established. Please check the text.
Line 214-221: What was the purpose of studying the effect of washing and sanitizing apples on the wild LAB population? This part does not appear to be related to this study, and the apples were not sanitized during the processing of Cider (as the author mentioned in Line 102). It is well known that the population of microorganisms must decrease after washing and sanitizing.
Answer: Thanks for the suggestion. The correlation between the first experiment and the others was improved. Please check lines 199-200 and 291-221.
Line 233: Here, the author says both ciders had the same yeast inoculum, with a population of 1.2 x 105 CFU/mL, which was not consistent with the method (Line 113-114).
Answer: Thank you for your comments. The values were checked and an average between the initial yeast populations was established. Please check the text.
Line 249: There are no error bars in all figures.
Answer: Thank you for your comments. Unfortunately, the error bars make it difficult to interpret the curves. In discussion with the other co-authors, we chose to keep the figures without the error bar.
Line 288-291: The results presented here for malic acid and lactic acid are inconsistent with those in Table 2. From Table 2, we can not see the difference in the malolactic bioconversion ability of the two kinds of Cider.
Answer: Thank you for your comments. The suggestion was accepted and changes were performed accordingly. Please check the information in text.
Line 331: For sensory evaluation, quantitative description of sensory features would be more effective as reported by Dravnieks (1985). The authors analyzed the volatile compounds in two different Ciders and showed that some compounds associated with floral and fruity aromas were significantly higher in CiderII, then what are the characteristics of CiderII through tasting?
Answer: Thank you for your comments. The suggestion was accepted and changes were performed accordingly. Please check the information in text.
Line 16: Saccharomyces cerevisiae can be abbreviated the second time it appears (same in L110), please check others. Line 56: Oenococcus oeni can be abbreviated the second time it appears (same in Line 111), please check others
Answer: Thank you for your comments. The suggestion was accepted and changes were performed accordingly. Please check the information in the text.
Line 127: “Ten milliliters”, please unify the unit format The unit of ℃ in the manuscript should be checked and revised, for most of them are messy.
Answer: Thank you for your comments. The suggestion was accepted and changes were performed accordingly. Please check the information in the text.
We think we were able to respond adequately to all issues raised by the reviewer and hope you will find our manuscript now acceptable for publication in Fermentation. If any further modifications are necessary, please inform us, and we will make every effort to implement them.
On behalf of all authors,
Yours sincerely,
Prof. Alessandro Nogueira,
State University of Ponta Grossa
Round 2
Reviewer 2 Report
Comments and Suggestions for Authors
-
Reviewer 3 Report
Comments and Suggestions for Authors
Accept in present form